# Water vapour dynamics as a key determinant of atmospheric composition and transport mechanisms

Andrew S. Kowalski[1,2], Ivan A. Janssens[3], Oscar Pérez-Priego[2,4]

[1]Department of Applied Physics, University of Granada, Granada, 18071, Spain
[2]Andalusian Institute for Earth System Research (IISTA), Granada, 18071, Spain
[3] Department of Biology, University of Antwerp; Wilrijk, Belgium
[4] Department of Forest Engineering, University of Córdoba; Córdoba, Spain

*Correspondence to*: Andrew S: Kowalski (andyk@ugr.es)

**Abstract.** Concentrations of any dry-air component (gas $i$) are often defined with reference to dry air, excluding water vapour.
Here it is shown that water vapour is an air component of paramount importance because its sources and sinks dominate those of air. Alone among atmospheric components, water vapor shifts from a trace constituent (practically negligible) to a bulk gas that meaningfully reduces gas $i$'s fraction and abundance within sultry, tropical air. Customary exclusion of humidity when expressing gas concentrations has practical justification, but biases assessments of gas $i$'s content within air. Overwhelmingly dominating surface exchanges, water vapor dynamics (WVD) influences gas $i$'s distributions, concentration gradients, and
transport mechanisms, but this has been overlooked due to reliance on gas fractions within dry air. Important implications of this include physical decoupling of leaf gas exchanges under very hot conditions, with extreme humidity inside stomatal pores acting physically to boost transpiration and simultaneously inhibit photosynthesis by suppressing carbon dioxide, consequences that ecology's stomatal conductance modelling framework has not elucidated. We accentuate the environmental significance of WVD and urge quantifying humidity whenever assessing fractional air composition.


# 1 Water Vapour is Part and Parcel of Air

Water vapour is often treated as separate from air due to its great variability, leading scientists to commonly quantify atmospheric composition in terms of fractions of dry air (Wallace and Hobbs, 2006). Such exclusion of humidity has practical justifications. For example, it avoids bias in chamber-based assessments of other gases' sources or sinks as a consequence of
evaporation (Pérez-Priego et al., 2015). Furthermore, air samples typically are dried before chemical analysis (Steele, 1987; Prinn et al., 1990; Keeling and Shertz, 1992; Novelli et al., 1999; Khalil et al., 2002; Tohjima et al., 2005) to prevent condensation and instrument damage. Thereafter, gas $i$'s dry-air fraction is commonly used to describe its distribution, concentration gradients, and thereby diffusive transport, and furthermore to assess its sources and sinks via inverse modelling (Bousquet et al., 1999; Tian et al., 2016). However, such discrimination—excluding water vapour, and often with no
accounting—disguises how water vapor dynamics (WVD) influences gas $i$'s abundance and gradients, and thereby obscures the hydrological cycle's role in actuating the mechanisms that transport gas $i$. Here we argue that water vapour is not apart from air—it *is* air—and specifically the component that most significantly influences gas abundances and transport dynamics, but these influences are masked by the typical exclusion of humidity when reporting on air composition.

The idea that water vapour is air is clearly demonstrated by ERA5 reanalysis data (Hersbach et al., 2020), selected with intent,
that contrast northern Spain's Atlantic and Mediterranean coasts near Bilbao and Barcelona, respectively. At 21:00 UTC on September 19th of 2023, sea-level air temperatures at these locations were essentially equal (21.6ºC), but the pressure ($p$) at the surface was lower in the Atlantic. According to the Ideal Gas Law, this means that between two identical bottles filled with ambient air, one from each coast, the Mediterranean container held more moles of air. Yet, due to higher humidity, it held fewer moles of dry air (lower partial pressure, $p_d$; Table 1). This clearly indicates that water vapour is air.


| Location | $p$ (hPa) | $T_d$ (ºC) | $e$ (hPa) | $p_d$ (hPa) |
|---|---|---|---|---|
| **Atlantic (43.75ºN, 2.5ºW)** | 1014 | 16.25 | 18 | 996 |
| **Mediterranean (40.25ºN, 1.5ºE)** | 1017 | 19.79 | 23 | 994 |

**Table 1. Comparison of atmospheric state at two points, respectively with atmospheric pressure ($p$) and dewpoint temperatures ($T_d$) from ERA5 reanalysis data (Hersbach et al., 2020) for September 19th, 2023, at 21:00 UTC. The vapour pressure ($e$) was calculated from $T_d$ using Eq. (10) of Bolton (1980), and the partial pressure of dry air ($p_d$) as the difference $p-e$.**


By dominating air's sources and sinks, water vapour uniquely exerts influence on atmospheric dynamics. Source–sink flows (Owen et al., 1985) are a type of fluid motion whose streamlines begin at sources and end at sinks. Though subtle in the atmosphere, such flows are shown below to contribute to transport, and they are largely governed by WVD. Since

evapotranspiration exceeds the combined surface fluxes of all dry-air components by orders of magnitude (Kowalski, 2017), to a very close approximation, air's sources and sinks are those of water vapor. This is reflected in the above example where the Mediterranean's higher sea-level $p$ is due to its greater humidity (Table 1), which in turn is forced by its superior evaporation rate (Lu, 2007). Thus, the substantial Atlantic–Mediterranean pressure gradient in Table 1 arose from WVD, and such gradients drive air motion (Sun et al., 2013). No other gas has comparable dynamical significance.

Net air movement causes non-diffusive transport that is distinct from diffusion in terms of its direction and determinants. At a given point, the orientation of non-diffusive transport—in the direction of fluid flow—is identical for every constituent of air. By contrast, diffusion inherently causes component gases to migrate in different directions. Also, the magnitude of non-diffusive transport depends on the air velocity and the abundance of gas $i$, but not directly on the concentration gradients that determine gas $i$'s diffusion. However, for certain gas components, their non-diffusive and diffusive transport mechanisms are both influenced by WVD and thereby intimately linked, but this is masked when referencing their abundances to dry air. Below, we describe how reliance on dry-air fractions that ignore non-negligible water vapour has restricted knowledge regarding gas abundances and transport mechanisms, and thereby important natural processes, at scales ranging from hemispheric (Keeling and Shertz, 1992) down to microscopic (Gaastra, 1959).

## 2 Humidity Suppresses Dry Air's Components

Neglect of WVD obscures how it significantly modulates the composition of air across climatic extremes. January's average sea-level isobars of 1008 hPa are found in both the Arctic and tropics (Ahrens, 2009). In surface environments with equal $p$, the partial pressure of water vapour ($e$)—tropically above 30 hPa but negligible in Arctic winters (Famiglietti et al., 2018)—regulates $p_d$ according to Dalton's law of partial pressures,

$$p = p_d + e. \tag{1}$$

Since $p_d$ is likewise the sum of dry air's component partial pressures ($\sum p_i$), this zero-sum game describes how extreme humidity suppresses every $p_i$ by several percent (Fig. 1). However, this effect is obscured when quantifying gas concentrations in relation to dry air, disguising how every gas $i$ is diluted and displaced by water vapour (Kowalski et al., 2021). Reported gas fractions that ignore humidity (Steele, 1987; Prinn et al., 1990; Keeling and Shertz, 1992; Novelli et al., 1999; Khalil et al., 2002; Tohjima et al., 2005) accurately portray dry air and sometimes help to identify gas $i$'s sources and sinks, but do not reflect the true concentrations (*i.e.*, including water vapour) that determine biogeochemical reactions. Also, for many dry-air components, WVD primarily determines the gradients that describe gas $i$'s diffusive transport via flux-gradient relationships such as Fick's 1$^{st}$ law.

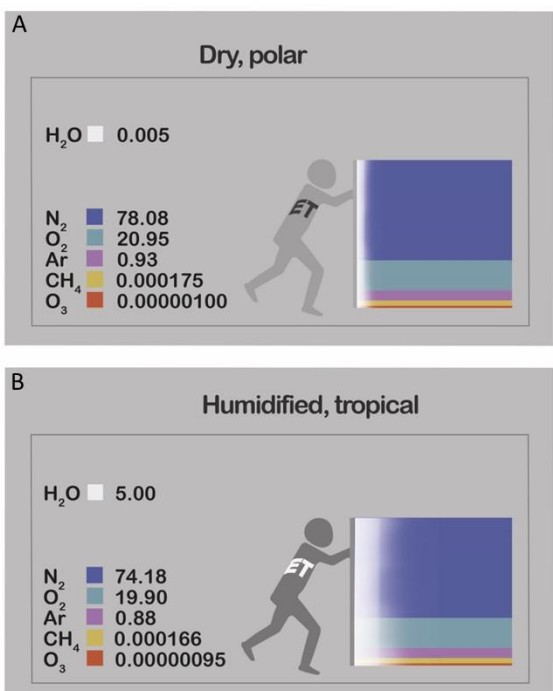

**Figure 1: Partial pressures (kPa) of water vapour (H₂O) and selected representatives of dry air – nitrogen (N₂), oxygen (O₂), argon (Ar), methane (CH₄) and ozone (O₃) – at 100 kPa pressure, comparing extremes of atmospheric humidity. Summing to 100 in each case, the values also represent molar fractions ($\chi_i$; %). (A) dry, polar (Wallace and Hobbs, 2006) versus (B) humidified to extreme sultriness (Raymond et al., 2020). Humidification's agent, evapotranspiration (ET) displaces dry air with water vapour. Illustration by Esther Cardell.**

Humidification can reduce $p_d$ by up to several percent, with varying degrees of influence on gas $i$'s gradients that delimit three categories of dry-air components (Kowalski and García-Valdecasas Ojeda, 2025), each typified in Fig. 1. Type I gases are scarce and highly reactive, like ozone whose $p_i$ varies over orders of magnitude (Robeson and Steyn, 1990) rendering negligible the influence of WVD. Type II gases like methane vary due to both their own sources and sinks (Knief, 2019) and also those of water vapour; methane's reactions modify its $p_i$ by a few percent (Steele et al., 1987), comparable in magnitude to forcing by humidity variations. Type III gases are inert relative to their abundance, and include dry air's every bulk component and noble gas; their $p_i$'s vary negligibly (far below 1%) due to their own sources and sinks, but substantially and inversely with humidity due to WVD.

Oxygen (O₂) is key example of a Type III gas whose overlooked humidity dependence is biogeochemically significant. Discarding water vapour before chemical analysis (Keeling and Shertz, 1992) confines O₂'s dry-air molar fraction ($c_i$) within 20.95+/-0.01%, i.e. varying globally by less than 100 ppm (Machta and Hughes, 1970). In reality, water vapour's molar fraction ($\chi_{H_2O}$) can reach 5% in extreme sultriness (Raymond et al., 2020), suppressing $\chi_i$—which better describes the true abundance

that determines biochemical or geophysical processes—by up to 10,000 ppm in the case of $O_2$ (Fig. 1). Thus, WVD makes $O_2$ variability orders of magnitude greater than previously supposed, with strong latitudinal and seasonal patterns in each hemisphere (Kowalski and García-Valdecasas Ojeda, 2025).

Geoscience studies that relied on dry-air $O_2$ fractions overlooked this. Meridional, oceanic $O_2$ transport was assessed (Portela et al., 2024) via a model that ignores latitudinal $p_i$ gradients when calculating dissolved $O_2$ (Aumont et al., 2015). Global $O_2$ cycle depictions claimed summer air is more aerobic (Gruber et al., 2001; Petsch, 2003), but humidity's strong seasonality makes the opposite true and much more so. Supposing $O_2$ to mirror carbon dioxide ($CO_2$; Gruber et al., 2001), hemispheric $O_2$ cycles were assumed to be decoupled (Keeling et al., 1998). But while $O_2$ reflects $CO_2$ regarding biochemical stoichiometry, the same is not true about physics. Abundances, gradients, and transport mechanisms of $CO_2$, which is a Type II gas, are fundamentally shaped by the carbon cycle and only secondarily influenced by WVD. By contrast, $O_2$ is a Type III gas that moves with dry air, whose cross-equatorial transport must be massive to buffer $p$ against seasonal shifts in $e$, per Eq. (1). For Type III gases like $O_2$, it is the hydrological cycle that overwhelmingly determines abundances, gradients, and transport mechanisms.

## 3 Transport Knowledge Gathers Steam

Assumptions that underly atmospheric transport models are invalidated by their neglect of WVD, wherein humidification dilutes gas $i$, induces its gradients, and activates its distinct transport mechanisms. Studies of transport that focus on $c_i$ (Bousquet et al., 1999) commonly suppose that production and consumption of gas $i$ directly determine its diffusion; these include "top-down" source/sink assessments of greenhouse gases based on inversion models (Tian et al., 2016). However, this overlooks how WVD drives gas $i$'s transport processes, as the following inertia-diffusion thought experiment illustrates. A steady-state tube contains moist air including static, inert dry air and water vapour migrating rightward from its source towards its sink (Fig. 2). This rightward migration yields mass flow—a net rightward air momentum and hence velocity. As noted in Section 1, air flows away from its sources and towards its sinks, which in Fig. 2 are exclusively those of water vapour. The resulting air velocity propels rightward, non-diffusive transport of every gas. But for gas $i$, leftward diffusion counters this since water vapour's source injects molecules that are not gas $i$, thereby diluting its concentration without directly altering its $c_i$. In Fig. 2, such displacement and dilution by water vapour activate offsetting transport mechanisms (Kowalski et al., 2021), yielding null net transport of gas $i$ via its upstream diffusion. Hence, analyses based solely on $c_i$ misrepresent the ongoing physical mechanisms of gas transport, particularly when water vapour exceeds trace concentrations (Kowalski, 2017), with significant implications for accurately representing diffusion.

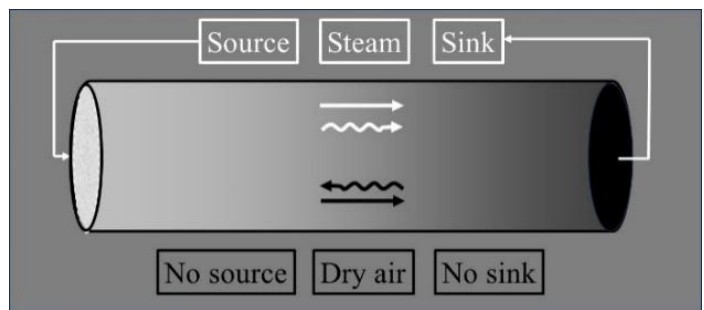

**Figure 2: A tube containing moist air, including water vapour (or steam, white) with a source on the left and a sink on the right, and nonreacting dry air (black). Greyscales depict gas fractions. Since air's momentum equals water vapour's, the air velocity is rightward. Both components experience rightward, non-diffusive transport (straight arrows) but they diffuse in opposite directions (curvy arrows). Thus, for dry air, offsetting transport mechanisms actuated by water vapour dynamics yield no net transport.**

Since WVD dominates atmospheric surface exchanges, the relative magnitudes of water vapour's transport mechanisms depend on its bulk/trace status, as extremes illustrate. At the moist extreme, if water vapour constitutes 99.9999% of air's mass in Fig. 2, its transport is primarily non-diffusive. In this case, trace dry air (1 mg kg$^{-1}$) has inertia that impedes air motion only negligibly, and water vapour's transport resembles simple source-to-sink airflow whose magnitude determines that of gas $i$'s upstream diffusion. This case—where air is water vapor—describes gas transport within a geyser, fumarole, or the spout of a whistling tea kettle (Fig. 3; $e \sim p$; $p_d \sim 0$). At the contrasting extreme, with mere trace water vapour, the fluid's bulk is dry air ($p_d \sim p$; $e \sim 0$) whose inertia keeps air nearly static, and water vapour transport is essentially diffusive; here, both airflow and dry-air diffusion are tiny, justifying traditional neglect of WVD when assessing gas $i$'s transport mechanisms for Type I and II gases. But this simplification fails for Type II gases when water vapour exceeds trace levels and hydrological forcing of transport mechanisms cannot be ignored, as is common in tropical climates and especially within the micro-scale humid pores that connect leaves to the atmosphere.

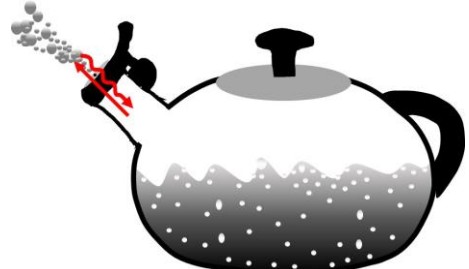

**Figure 3: A whistling tea kettle, containing pure water vapour both above and within liquid water (as steam bubbles). At the boiling point, the water vapour pressure equals the total pressure. Water vapour egress is non-diffusive, with the wet steam jet visible due to external condensation. Inward diffusion of dry-air component gas $i$ (curvy line) is huge,**

**due to an extreme concentration gradient with no gas *i* inside the boiling kettle, but insufficient to overcome its outbound, non-diffusive transport (straight line).**

## 4 Consequences of Bulk Water Vapour within Stomata

Perhaps the most significant yet overlooked consequences of WVD occur in the micro-scale environments of plant stomata, where at high temperatures its neglect invalidates foundational ecological models. Inside terrestrial plant leaves, water vapour emission from humidified microscopic cavities beneath stomatal pores (i.e., transpiration; Buckley and Sack, 2017) vastly exceeds photosynthetic $CO_2$ and $O_2$ fluxes. Traditionally, plant physiologists have modelled stomatal gas transport as purely diffusive (Moss and Rawlins, 1963), regulated by stomatal conductance ($g_s$). This assumes proportional $CO_2$ and water vapour flux/gradient ratios based on Graham's law (Jones, 2014) and allows estimating the dry-air molar $CO_2$ fraction ($c_i$) of substomatal cavities, presumed a suitable proxy for the $p_i$ of $CO_2$ (Gaastra, 1959), which determines $CO_2$ dissolution via Henry's law and thereby photosynthesis. Such reasoning forms the basis of stomatal optimality theory (Cowan and Farquhar, 1977), which posits that plants adjust $g_s$ to optimise carbon gain relative to water loss, thus maximising water-use efficiency (WUE) via proportional regulation of coupled gas exchanges. However, this paradigm neglects both $CO_2$ suppression by humidification (Section 2) and non-diffusive transport (Section 3), such that the validity of model-laden parameters $c_i$ and $g_s$ hinges on the assumption that water vapour is a trace gas. As discussed below, such neglect of WVD is inappropriate at very high leaf temperatures (Kowalski, 2025).

Water vapour's trace or bulk status can be quantified using its mass fraction, known as the specific humidity ($q$). Because it relates to inertia (Section 3), $q$ also defines the non-diffusive fraction of water vapour transport, as described by momentum conservation (Kowalski, 2017; see Appendix A). Inside sub-stomatal cavities, $q$ reaches non-negligible levels over relevant ranges of leaf air's dew point temperature and $p$ (Fig. 4A).

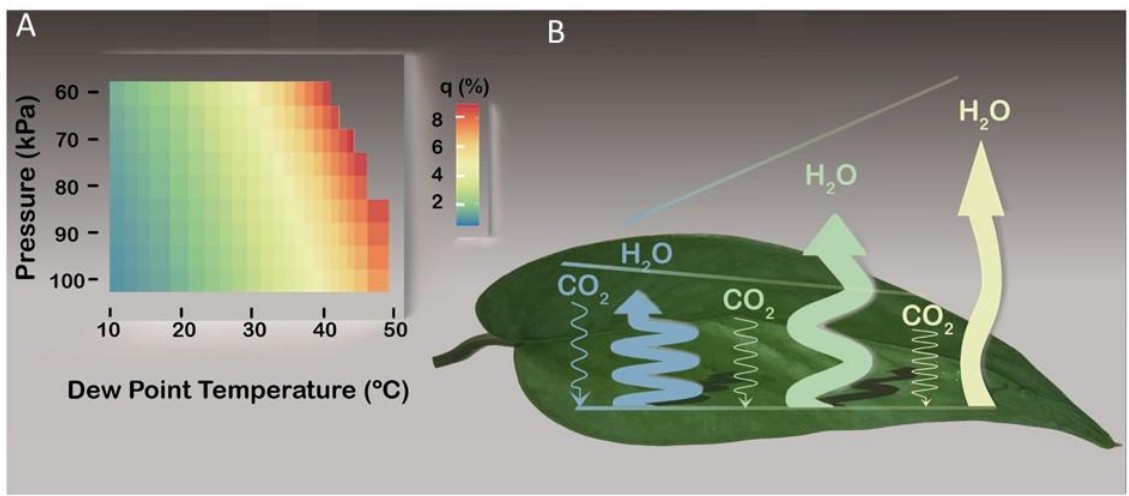

**Figure 4: The non-diffusive fraction of water vapour ($H_2O$) transport, or specific humidity ($q$; see Appendix A), and how it decouples leaf gas exchanges. (A) Colour map of stomatal $q$, calculated from leaf air's dew point temperature and pressure (Petty, 2008). (B) Schematic of $CO_2$ and $H_2O$ fluxes decoupling as $q$ increases (blue-yellow transition). Flux magnitudes are represented by arrow widths ($H_2O$ far exceeding $CO_2$) and lengths (not to scale) that depict how elevated $q$ boosts $H_2O$ egress but retards $CO_2$ ingress, despite increased inward $CO_2$ diffusion. Line sinuosity depicts the degree to which transport is diffusive. Illustration by Esther Cardell.**

Diffusion-only stomatal transport models, including ternary approaches (Jarman, 1974; von Caemmerer and Farquhar, 1981), obscure two key consequences of WVD when humidity achieves bulk status. First but least, since $q$ varies inversely with $p$ (Fig. 4A), leaf functioning depends on elevation in contradiction to deductions based on diffusion-only models (Gale, 1972). This may help explain why vascular plants that evolved *without* stomata thrived only in regions with high-$q$ leaves—in tropical latitudes at very high altitude (Keeley et al., 1984)—when considering the second consequence. To wit, mass flow indiscriminately drives both $CO_2$ and water vapour out of the leaf (Section 3), which reduces WUE via independent effects on each gas. Within open stomata at high $T$ when $q$ is high (Fig. 4a) and independently of dilation status, bulk water vapour suppresses the $\chi_i$ of every dry-air component (Fig. 1), including $CO_2$, thereby reducing net assimilation ($A_n$) without changing $c_i$; meanwhile, WVD enhances water vapour egress (Fig. 4B) via non-diffusive transport. Evidence of such decoupling—with extreme heat physically restraining $A_n$ but boosting ET—has frequently been observed across a range of ecosystems and species, as reviewed below.

In the Sonoran Desert with leaf $T$ reaching $50\,°C$, 37% of surveyed species showed increased ET despite declining $A_n$ (Aparecido et al., 2020). For broadleaf species during heatwaves under well-watered conditions, Diao et al. (2024) found that

$A_n$ decreased with rising heat while ET increased continuously up to a maximum cuvette T of 40°C. Marchin et al. (2023) found that heatwaves in urban Sydney caused broadleaf evergreen and deciduous species to increase $g_s$ but decrease $A_n$ at air T above 45ºC, independent of plant water access. At the ecosystem scale, Krich et al. (2022) observed similar decoupling in four Australian woodland species based on eddy covariance fluxes, and De Kauwe et al. (2019) demonstrated widespread model–data mismatches during heat extremes across FLUXNET sites. All of these findings are consistent with WVD being a

major driving force that physically decouples the fluxes of these gases and reduces WUE in leaves at high T.

     To illustrate the significance of WVD's influence on high-temperature leaves, we incorporated the consequences of Dalton's law into the model of Scafaro et al. (2023). These authors showed that biochemistry can limit photosynthesis when leaf T rises above 40 °C; however, by making the substomatal $p_i$ of $CO_2$ a fixed fraction of ambient, they effectively bypassed WVD's suppression of dry air within these extremely humid leaf airspaces. To correct this, we multiplied each $p_i$ by $(1 - \chi_{H_2O})$ in the

Scafaro et al. dataset, which improved the fit of the photosynthetic T response function and reduced bias in Rubisco carboxylation–limited assimilation rates (Fig. 5).

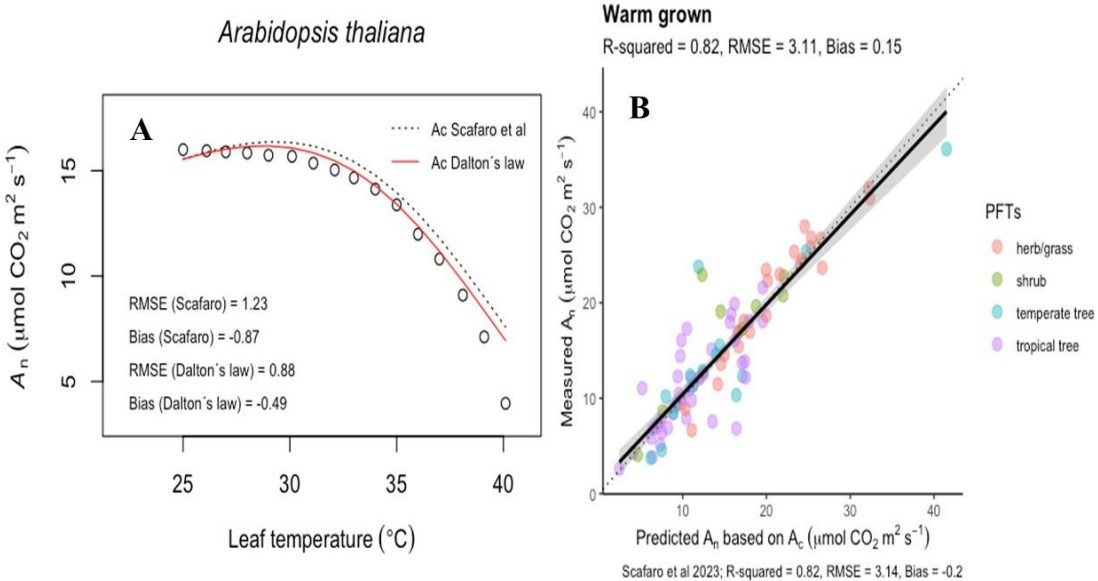

**Figure 5: Observations, modelled simulations, and the predictability of the leaf temperature response of net $CO_2$ assimilation ($A_n$). (A) The $A_n$ temperature response of *Arabidopsis Thaliana*. As in Scafaro et al. (2023), points are**

**observations and the curves are the modelled Rubisco carboxylation limited assimilation rates ($A_c$) with Rubisco deactivation included. Model partial pressures are estimated via the dry-air molar fraction ($c_i$, dashed line) or moist-air molar fraction ($\chi_i$; solid red line). (B) Predictions of $A_c$ which included Rubisco deactivation but with partial pressures estimated via $\chi_i$.**

## 5 Perspectives and Conclusions

At each spatial scale considered above, disregard of WVD emerges as a substantial shortcoming that results from reliance on dry-air fractions. This highlights the importance of accounting for humidity in assessments of gas $i$'s abundance and transport mechanisms, except perhaps for Type I gases or studies confined to extremely dry environments (e.g., Antarctic science). Fortunately, analytical limitations to determining moist air's composition can be compensated by simple conversion of dry-air fractions to air fractions. The conversion factor, obtained dividing gas $i$'s molar fraction $\left(\chi_i = \frac{p_i}{p}\right)$ by its dry-air molar fraction $\left(c_i = \frac{p_i}{p_d}\right)$ and substituting via Eq. (1), is

$$\frac{\chi_i}{c_i} = \frac{p_d}{p} = \frac{p-e}{p} = 1 - \chi_{H_2O}, \tag{2}$$

Figures 1 and 5 quantitatively characterise the effects of this conversion. Mass conservation similarly allows calculating gas $i$'s mass fraction from its dry-air mass fraction (or mixing ratio) and $q$. Enabling such conversions requires only inexpensive instruments—a barometer and thermo-hygrometer—to monitor the state and humidity of sampled air.

By unlocking the potential of WVD, the steam engine powered the industrial revolution (Nuvolari, 2006) and influenced economic, social and political systems worldwide. Modern environmental science should similarly harness a renewed appreciation of WVD's consequences to advance insight in atmospheric and ecological sciences. The above instances——key examples where its neglect has impeded understanding of the natural world——underline the relevance of quantifying humidity when describing gas concentrations and transport mechanisms. Regarding plant ecology, a novel modelling framework seems necessary to accurately quantify how high temperatures, particularly at altitude, decouple leaf photosynthesis and transpiration.

## Competing interests

The authors declare that they have no competing interests.

## Author contributions

ASK conceived and wrote the first draft of the manuscript, created Figures 2 and 3, and contracted for figure artwork. IAJ reoriented key sections, notably the abstract and first section, and eliminated much extraneous text. OP-P wrote the first-draft paragraph on leaf ecology, produced Figures 4A and 5, and remoulded the conclusions. All authors revised the manuscript.

**Acknowledgements**

ASK and OPP are funded by the *Ministerio de Ciencia e Innovación* project REMEDIO (grant no. PID2021-128463OB-I00).
IAJ acknowledges support from ICOS-Flanders (Grant No I00325N). We thank Matilde García-Valdecasas Ojeda for
assistance in accessing ERA5 reanalysis data, editor Lutz Merbold for handling and suggesting changes to our manuscript, and
Dan Yakir and one anonymous referee for reviewing our submission. We used AI tools to improve paragraph concision.

**Appendix A - Stomatal Water Vapour Transport's Non-diffusive Fraction is the Specific Humidity ($q$)**

For most land plants, the gaseous species composing stomatal air have diverse momenta (kg m s$^{-1}$) including outward water
vapour ($H_2O$) and $O_2$, null nitrogen ($N_2$) and argon (Ar), and inward $CO_2$. With the outward velocity as $w$ (m s$^{-1}$) and the mass
as $m$ (kg), air's momentum is the sum of its components' momenta

$$wm = (wm)_{N_2} + (wm)_{O_2} + (wm)_{Ar} + (wm)_{H_2O} + (wm)_{CO_2} + \dots \tag{A1}$$

Dividing each side of Eq. (A1) by some stomatal control volume $V$ yields

$$w\rho = (w\rho)_{N_2} + (w\rho)_{O_2} + (w\rho)_{Ar} + (w\rho)_{H_2O} + (w\rho)_{CO_2} + \cdots, \tag{A2}$$

where $\rho$ is density (kg m$^{-3}$) and $(w\rho)_i$ is the flux density (kg m$^{-2}$ s$^{-1}$) of gas $i$. Since $H_2O$ dominates stomatal gas exchange by
orders of magnitude (Kowalski, 2017), the flux densities of other gases can be neglected simplifying Eq. (A2) to

$$w\rho = (w\rho)_{H_2O}, \tag{A3}$$

which quantifies the net $H_2O$ flux density, or evaporative flux density ($E$; kg m$^{-2}$ s$^{-1}$). Dividing this by $\rho$ yields the air velocity

$$w = \frac{E}{\rho} \tag{A4}$$

and multiplying this by $\rho_{H_2O}$ defines $H_2O$'s non-diffusive flux density ($F_{non,H_2O}$) as

$$F_{non,H_2O} = \frac{E}{\rho}\rho_{H_2O}, \tag{A5}$$

where $\frac{\rho_{H_2O}}{\rho}$ (the specific humidity, $q$) quantifies the non-diffusive fraction of $H_2O$ transport, as depicted in Figure 4.

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
