# Peer review of "Water vapour dynamics as a key determinant of atmospheric composition and transport mechanisms"

_EGUsphere, 2025_

## Author Comment (AC1)

*Review of Kowalski et al., Water vapor dynamics…*

**This Viewpoint paper offers a provocative perspective of the role of water vapor in atmospheric and leaf-scale gas exchange. It argues that water vapor dynamics (WVD) can be a driver of gas transport phenomena, using first-principles reasoning and thought experiments. The paper challenges the conventional (and useful) practice that expresses gas concentrations relative to dry air, especially under very humid conditions. It raises valid conceptual challenges to current modeling frameworks in both atmospheric chemistry and plant ecophysiology.**

We thank Dr. Yakir for this considered assessment. Clearly the conventional practice of expressing gas fractions relative to dry air is often useful (lines 24-25 of our manuscript explicitly recognize this), and the stomatal conductance modelling framework successfully describes the coupled exchanges of water vapor and carbon dioxide within a broad range of environmental conditions. However, while this encompasses most growth conditions for terrestrial plants, it includes neither vital gas exchanges at very high leaf temperatures nor the transport of oxygen (a type III gas) in general. The point of our manuscript is to extend descriptions of gas concentrations and transport mechanisms to situations that conventional practice does not correctly describe.

As Dr. Yakir recognizes, our manuscript is about gas physics. Although we claim that this has implications that extend to plant ecophysiology, we prefer not to expand the scope of the paper to make it predominantly about leaf functioning during heatwaves. We hope that the reviewer(s) and editor will take this into account when considering our replies below.

**However, the discussion of using mole fraction and partial pressure of dry air is not new, and the cases raised here apply mostly to extreme and rare cases, and remain speculative due to limited direct empirical validation. Some of the claims may be too strong like the relevance WVD in driving bulk airflow in atmospheric boundary layer dynamics. Or the suggestion of widespread invalidation of top-down flux inversion models (without demonstrating practical model biases attributable to neglecting WVD).**

We think our discussion is new, but are eager to learn about previous studies that discuss the consequences of Dalton's law of partial pressures in this context.

Our paper is based on physics, not speculation, and direct empirical validation is readily available in literature data when taking Dalton's law into account. As an explicit example, practitioners of the Bowen ratio method (Perez et al., 1999; Savage et al., 2009) have characterised representative vertical water vapour pressure gradients ($\frac{de}{dz}$) as -100 Pa m$^{-1}$ over actively transpiring vegetation. This decrease far exceeds the vertical pressure decline from the hydrostatic equation ($\frac{dp}{dz} = -\rho g$), which is only about -10 Pa m$^{-1}$. When taking the derivative of Dalton's law—our Eq. (1)—with respect to height

$$\frac{dp}{dz} = \frac{dp_d}{dz} + \frac{de}{dz}$$

these empirical data indicate that the dry-air pressure must *increase* with height ($\frac{dp_d}{dz} > 0$), implying that the oxygen concentration increases with height and therefore that oxygen diffusion is downward due to water vapor dynamics (see example in Table 1).

| $z$ (m) | $p$ (Pa) | $e$ (Pa) | $p_d$ (Pa) | $p_{N2}$ (Pa) | $p_{O2}$ (Pa) | $p_{Ar}$ (Pa) | $\chi_{O2}$ (ppt) |
|---------|----------|----------|------------|---------------|---------------|---------------|-------------------|
| 0.8 | 100010 | 2100 | 97910 | 76468 | 20561 | 881 | 205.6 |
| 1.8 | 100000 | 2000 | 98000 | 76538 | 20580 | 882 | 205.8 |

**Table 1. An example of the evolution with height ($z$) of atmospheric pressure ($p$), the partial pressures of water vapor ($e$), dry air ($p_d$), nitrogen ($p_{N2}$), oxygen ($p_{O2}$), and argon ($p_{Ar}$), and the molar fraction of oxygen ($\chi_{O2}$) over short, transpiring vegetation. Dry air's partial pressure is derived from Dalton's law, and its composition supposes every 1000 molecules to include 781 molecules of nitrogen, 210 of oxygen, and 9 of argon.**

Since transpiration reduces near-surface oxygen by 200 ppm (0.2 ppt; Table 1) versus aloft, diffusion of oxygen (a Type III gas) is downward regardless of photosynthetic production. Yet we know that *net* oxygen transport over a photosynthetic surface is upward, making it clear that the vertical flux of oxygen from the hypoxic, near-surface layer towards higher, more aerobic regions of the boundary layer is non-diffusive in nature and depends on bulk airflow driven by water vapor dynamics.

Finally, our manuscript suggests that what is invalidated by neglect of water vapour dynamics is not necessarily top-down flux inversion models, but only their underlying assumptions. It is not inconceivable that such models could correctly describe sources and sinks despite mischaracterising the physical mechanisms of transport (e.g., see Yan et al., 2023 and especially the on-line discussion phase of that paper). However, this is unlikely to be true in tropical conditions where water vapor surpasses trace status.

References:

Perez, P. J., et al., 1999, Assessment of reliability of Bowen ratio method for partitioning fluxes, *Agricultural and Forest Meteorology*, **97**, 141–150.

Savage, M.J., et al., 2009, Bowen ratio evaporation measurement in a remote montane grassland: Data integrity and fluxes, *Journal of Hydrology*, **376**, 249–260, https://doi.org/10.1016/j.jhydrol.2009.07.038

Yan, Y. et al., 2023, A modeling approach to investigate drivers, variability and uncertainties in $O_2$ fluxes and $O_2 : CO_2$ exchange ratios in a temperate forest, *Biogeosciences*, **20**, 4087–4107, https://doi.org/10.5194/bg-20-4087-2023

*The authors use sound theoretical reasoning to develop the WVD idea but extending these principles into stomatal decoupling, and macro-scale transport, relies on indirect support, and many of the supporting studies (e.g., Kowalski 2017; Kowalski et al., 2021, 2025) are from the same authors.*

We agree with the referee's comment. The support, apart from the sound physics, is indeed indirect, but this does not imply it is incorrect. The hypoxic nature of very humid air is a direct consequence of Dalton's law of partial pressures when applied in an approximately isobaric context, as in the table/model above and also our Figure 1. Oxygen is consistently transported up its gradient from relatively hypoxic regions—where it is produced yet diluted by water vapor dynamics—toward drier, more aerobic regions. Such transport is independent of scale:

- From substomatal cavities to the exterior leaf environment (Kowalski, 2025);
- from near-surface air to higher regions of the boundary layer (Table 1, above); and
- from tropical rainforests to other areas of the world (Figure 1 of our manuscript).

In each case, net oxygen transport is oriented not by diffusion but rather bulk airflow. This makes clear that water vapor dynamics actuates both types of transport mechanisms.

*Similarly, the idea that WVD, under high humidity and temperature, could lead to a physical decoupling of $CO_2$ uptake and $H_2O$ loss in leaf gas exchange is intriguing. But validation with field or chamber measurements is badly missing.*

Here we respectfully disagree with the referee. Numerous field and chamber measurements (Aparecido et al., 2020; de Kauwe et al., 2019; Diao et al., 2024; Krich et al., 2022; Marchin et al., 2023) have observed decoupling under heatwave conditions. Further validation is available via our re-analysis of the results of Scafaro et al. (2023) dataset. We confirmed via extensive personal communication with the authors (both Drs. Farquhar and Scafaro) that their model imposed constant values of $p_{CO_2}$ and $p_{O_2}$ even at leaf temperatures far exceeding 40ºC, thus short-circuiting dilution by water vapor. We multiplied these partial pressures by $\left(1 - \chi_{H_2O}\right)$ within the Scafaro et al. (2023) dataset—effectively correcting for the effects of dilution— and this improved the fit to observations of an assimilation temperature response function, and also reduced bias in Rubisco carboxylation limited assimilation rates (Fig. 1). Please, note that Scafaro´s dataset, as well as most existing studies, rarely contain field and chamber measurements exceeding 40ºC when the effects of decoupling are relevant, and the model bias remarkable (See Fig. 1).

[Figure]

**Figure 1. Observations, modelled simulations, and the predictability of the leaf temperature response of net CO₂ assimilation ($A_n$). A** The $A_n$ temperature response of *Arabidopsis Thaliana*. As in Scafaro et al.[1], points are observations and the curves are the modelled Rubisco carboxylation limited assimilation rates ($A_c$) with Rubisco deactivation included. Model partial pressures are estimated via the dry-air molar fraction ($c$, dashed line) or moist-air molar fraction ($\chi$; solid red line). **B** Predictions of $A_c$ which included Rubisco deactivation but with partial pressures estimated via the moist-air molar fraction ($\chi$; dashed line).

We prefer not to include the results shown in Figure 1 above in this paper that focuses on atmospheric composition and transport mechanisms. Should we succeed in publishing this paper regarding the physics, we will then aspire to publish a follow-up manuscript that specifically regards leaf functioning, and will include field or chamber measurements. However, if the referees and editor feel that their inclusion is necessary, then we will be willing to oblige.

*In fact, the widely used model of Farquhar et al 1980 deals with some of the aspects of CO2 dilution by water vapor in the substomatal space, but is not cited or discussed. in fact, all leaf gas exchange measurements are also corrected for humidity dilution in calculating net assimilation.*

Farquhar et al (1980) make no mention, neither of water vapor, humidity, nor dilution. We think the paper Dr. Yakir has in mind is that of von Caemmerer and Farquhar (1981), which we did cite and which indeed does account for dilution of carbon dioxide by water vapor.

However, there are two aspects of this paper that are criticizable and upon which we think our own manuscript improves:

1. Because dilution by water vapor is hidden in an appendix of their paper, it has not generally been recognized by ecophysiologists. This likely explains why high-temperature suppression of photosynthesis has been attributed solely to biochemical processes (as we note at line 171) and may even explain why dilution was excluded from the model of Scafaro et al. (2023), even with Dr. Farquhar as a co-author. Hence our critical remark at line 165 (such approaches "obscure two key consequences of water vapor dynamics when humidity achieves bulk status"); and

2. The description of diffusion based on gradients in the mole fraction (Jarman, 1974; von Caemmerer and Farquhar, 1981) represents a violation of Newton's laws. This is demonstrated both in a previous publication (Kowalski et al., 2021) and in two open discussions that can be accessed here:
   a. https://doi.org/10.5194/egusphere-2025-2814
   b. https://doi.org/10.5194/egusphere-2025-2705-RC1

   Rather, because of the key role played by mass/inertia in defining motion, it is the mass fraction whose gradients determine diffusion and which must be used to distinguish diffusive from non-diffusive transport (Kowalski, 2017; see also the Appendix of our manuscript). The von Caemmerer and Farquhar (1981) paper does not recognize non-diffusive transport by bulk airflow exiting stomata. While this mischaracterizes the physics of leaf gas exchanges with the atmosphere, in most studies its effect would be negligible. However, when studying impacts of very high temperature on gas transport, ignoring or misrepresenting water vapor dynamics does matter.

Reference:

Farquhar, G.D., von Caemmerer, S. and Berry, J. A., 1980, A biochemical model of photosynthetic $CO_2$ assimilation in leaves of C3 species, *Planta*, **149**, 78–90.

***In discussing the links to biochemical rates, specifically in photosynthesis, some reference should be made to the fact that dissolved CO2 is the end member via Henry's law and other local factors at the site.***

We agree. Therefore, we propose to change the sentence that begins at line 145 from

- "This assumes proportional $CO_2$ and water vapour flux/gradient ratios based on Graham's law (Jones, 2014) and allows estimating the dry-air molar $CO_2$ fraction ($c_i$) of substomatal cavities, presumed a suitable proxy for the $p_i$ of $CO_2$ (Gaastra, 1959), which is a key photosynthetic determinant", to

- "This assumes proportional $CO_2$ and water vapour flux/gradient ratios based on Graham's law (Jones, 2014) and allows estimating the dry-air molar $CO_2$ fraction

($c_i$) of substomatal cavities, presumed a suitable proxy for the $p_i$ of $CO_2$ (Gaastra, 1959), which determines $CO_2$ dissolution via Henry's law and thereby photosynthesis".

**Rhetoric like "air is water vapor; water vapor is air" is somewhat distracting**

We see the referee's point, but still believe this rhetoric is important to get the reader's attention. Grasping these ideas is essential to understanding the influence of water vapor dynamics on bulk airflow. Staying with the order of scales that is presented above, we note that:

- By pumping air into substomatal cavities, water vapor dynamics force bulk airflow out of stomata that enables up-gradient oxygen transport and, at very high temperatures, becomes relevant to the transport of both water vapor and carbon dioxide;
- By pumping air out of the surface, water vapor dynamics push the entire boundary layer upward and enables upward (up-gradient) transport of oxygen; and
- By pumping massive amounts of air into tropical rainforests, water vapor dynamics push air away from them and allows them to export oxygen to more aerobic regions.

**The schematic showing decoupled $CO_2$ and $H_2O$ fluxes is conceptually useful but would benefit from real data overlay (e.g., from gas exchange measurements during heatwaves) to illustrate feasibility.**

Thank you for this comment, but observations of decoupled carbon dioxide and water vapor exchanges during heatwaves have been broadly observed by plant ecophysiologists (Aparecido et al., 2020; de Kauwe et al., 2019; Diao et al., 2024; Krich et al., 2022; Marchin et al., 2023). Here, we prefer to focus on the roles of atmospheric composition and transport mechanisms, not on the ecophysiology. While referring to these consequences for ecophysiology during heatwaves as observed in many recent studies, we therefore prefer not to include this in this manuscript.

**Equations A5, and 2, could benefit from some relevant quantitative examples.**

We thank the referee for this suggestion and intend to add statements leading the readers to the quantitative examples in the manuscript. Thus, we propose to add

- after Equation 2 that a quantitative example of this is depicted in Figure 1; and
- a final sentence to the Appendix stating that the consequences of Equation A5 are visualized in Figure 4.

*Overall, the manuscript presents a thought-provoking argument that challenges long-standing assumptions in atmospheric and plant sciences. It presents a sound physical and conceptual relevance of water vapor as an important player in gas transport and exchange. It would benefit from: Stronger empirical support through simulations or re-analysis of published data. Moderation of rhetoric in places where established practices are critiqued.*

We sincerely thank the referee for the time and work invested in our study. We hope that the discussion above convinces the referee that there is ample empirical support in the published literature. If we were to include empirical data in this manuscript, the focus would shift from the underlying physics to the ecophysiological consequences, which we really want to avoid in this manuscript.

Any specific suggestions for moderation of rhetoric would be most welcome.

*I think the manuscript should be accepted for publication after moderate revision, particularly with a better balance of conceptual divergence with empirical grounding.*

Again, we thank Dr. Yakir for his careful examination of our paper.

---

## Author Comment (AC3)

**Summary**

The authors outline current shortcomings in this discussion paper from excluding the impacts of water vapour by the use of dry air equivalents. In certain cases, this may impede understanding of measurements or cause inappropriate model outcomes especially for ecosystem-atmosphere interactions. Using thought-experiments, they discuss this issue, its effects and call to improve such models as well as use mole fractions to include water vapour.

We thank the reviewer for taking the time to assess our preprint and for providing helpful comments. The above paragraph summarizes our paper adequately, and we believe the suggestions below will help us to improve the paper upon revision.

There indeed seem to be cases where the outlined issues occur, which could be further highlighted with specific examples. However, many of these cases are only found in very specific or extreme conditions. Furthermore, the current availability of observational data from such cases are still few, requiring theoretical considerations and some speculation as to the magnitude and frequency of their impacts. The inclusion of further supporting data and concrete examples using measurements would be helpful to illustrate the authors arguments.

We agree that concrete examples are needed. Indeed, in our preprint the sentence at lines 61-63 presaged evidence for hemispheric consequences but the paragraph regarding oxygen (beginning at line 94) came up short in this regard. Therefore, we propose to replace that paragraph with the following two:

Oxygen (O2) is key example of a Type III gas whose overlooked humidity dependence is biogeochemically significant. Discarding water vapour before chemical analysis (Keeling and Shertz, 1992) confines O2's dry-air molar fraction ( $c_i$ ) within 20.95+/-0.01%, i.e. varying globally by less than 100 ppm (Machta and Hughes, 1970). In reality, water vapour's molar fraction ( $\chi_{\rm H_2O}$ ) can reach 5% in extreme sultriness (Raymond et al., 2020), suppressing  $\chi_i$ —which better describes the true abundance that determines biochemical and geophysical processes—by up to 10,000 ppm in the case of O2 (Fig. 1). Thus, WVD makes O2 variability orders of magnitude greater than previously supposed, with strong latitudinal and seasonal patterns in each hemisphere (Kowalski and García- Valdecasas Ojeda, 2025).

Geoscience studies that relied on dry-air O2 fractions overlooked this. Meridional, oceanic O2 transport was assessed (Portela et al., 2024) via a model that ignores latitudinal  $p_i$  gradients when calculating dissolved O2 (Aumont et al., 2015). Global O2 cycle depictions claimed summer air is more aerobic (Gruber et al., 2001; Petsch, 2003), but humidity's strong seasonality makes the opposite true and much more so. Supposing O2 to mirror carbon

dioxide (CO2; Gruber et al., 2001), hemispheric O2 cycles were assumed to be decoupled (Keeling et al., 1998). But while O2 reflects CO2 regarding biochemical stoichiometry, the same is not true about physics. Abundances, gradients, and transport mechanisms of CO2, which is a Type II gas, are fundamentally shaped by the carbon cycle and only secondarily influenced by WVD. By contrast, O2 is a Type III gas that moves with dry air, whose crossequatorial transport must be massive to buffer *p* against seasonal shifts in *e*, per Eq. (1). For Type III gases like O2, it is the hydrological cycle that overwhelmingly determines abundances, gradients, and transport mechanisms.

**Additional References:**

Aumont, O. et al., PISCES-v2: an ocean biogeochemical model for carbon and ecosystem studies, *Geosci. Model Devel.*, **8**, 2465–2513, <a href="https://doi.org/10.5194/gmd-8-2465-2015">https://doi.org/10.5194/gmd-8-2465-2015</a>, 2015.

Gruber, N., M. et al., Air-sea flux of oxygen estimated from bulk data: Implications For the marine and atmospheric oxygen cycles, *Global Biogeochem*. *Cycles*, **15**(4), 783–803, <a href="https://doi.org/10.1029/2000GB001302">https://doi.org/10.1029/2000GB001302</a>, 2001.

Keeling, R. F. et al., Seasonal variations in the atmospheric O2/N2 ratio in relation to the kinetics of air-sea gas exchange, *Global Biogeochem*. *Cycles*, **12**, 141-163, <a href="https://doi.org/10.1029/97GB02339">https://doi.org/10.1029/97GB02339</a>, 1998.

Petsch, S. T., The global oxygen cycle. In Schlesinger, W. et al., Treatise on Geochemistry Volume 8, Amsterdam: Elsevier, 2003.

Portela, E. et al., The ocean's meridional oxygen transport. *J. Geophys. Res.: Oceans*, **129**, e2023JC020259 <a href="https://doi.org/10.1029/2023JC020259">https://doi.org/10.1029/2023JC020259</a>, 2024.

Overall, the paper acts as an incentive to increase data collection of such cases, as well as creating awareness for the current model shortcomings, which the community need to consider when evaluating their own data. The paper should be accepted after some revision and linguistic editing.

We thank the reviewer for this encouraging assessment.

**Specific points**

1. Generally, a well-conceived idea to raise awareness of such issues. That is why basing arguments on concepts of "air is water vapour" detracts from the overall discussion and aim. Because water vapour can in some cases reach bulk concentrations, does not warrant it to assume parity with the main constituents in

all cases. Naturally, the addition of a gas will change the relative distribution among the other constituents. The discussion could be dedicated less to this effect and more to its impacts.

Two reviewers have objected to the phrase "air is water vapor," indicating the need to reword our narrative in this regard. We hope that the following revisions express the underlying idea with greater clarity:

- a. In the abstract, we propose change the offending sentence to: "Here it is shown that water vapour is an air component of paramount importance because its sources and sinks dominate those of air."; and
- b. We propose to substitute the paragraph that begins at line 46 with the following:

By dominating air's sources and sinks, water vapour uniquely exerts influence on atmospheric dynamics. Source—sink flows (Owen et al., 1985) are a type of fluid motion whose streamlines begin at sources and end at sinks. Though subtle in the atmosphere, such flows are shown below to contribute to transport, and they are largely governed by WVD. Since evapotranspiration exceeds the combined surface fluxes of all dry-air components by orders of magnitude (Kowalski, 2017), to a very close approximation, air's sources and sinks are those of water vapor. This is reflected in the above example where the Mediterranean's higher sea-level *p* is due to its greater humidity (Table 1), which in turn is forced by its superior evaporation rate (Lu, 2007). Thus, the substantial Atlantic—Mediterranean pressure gradient in Table 1 arose from WVD, and such gradients drive air motion (Sun et al., 2013). No other gas has comparable dynamical significance.

2. The authors omit any discussion considering humidity corrections in observational data to account for the impacts of water vapour in the aforementioned cases and references to relevant method papers or examples where such effects were observed could be added.

We think that the paragraph proposed above for insertion into the manuscript regarding oxygen provides several examples that show the benefits of correctly accounting for humidity. Our equation (2) makes the relevant humidity corrections, and as suggested by the first chronological reviewer (Dr. Yakir) we propose to add a comment following that equation to note that a quantitative example of this is depicted in Figure 1.

3. Replacing dry air measures simply with molar fractions that include water vapour would likely cause more issues than it would solve, except in the few circumstances outlined in the manuscript in which cases molar fractions could be

used. Also, this method would remove this useful measure for independent comparisons. If decoupling of CO2 and H2O at stomatal interfaces has been observed due to this effect, then there should be data available to model these processes with the improved approach suggested by the authors to illustrate the importance of such effects.

We do not propose abandoning dry-air measures, but we do call for ending the systematic neglect of humidity. Constant inclusion of humidity data when monitoring and reporting gas concentrations would allow researchers to account for water vapour fluctuations where relevant. The paragraph proposed above for insertion into the manuscript regarding oxygen provides several examples where reliance on dry-air fractions has restricted knowledge. Regarding the stomatal gas exchanges, as noted in our reply to Dr. Yakir, we prefer to keep the focus of this paper on the underlying physics and not make it largely about ecophysiology. We agree that a new modeling framework is needed for high-temperature stomatal gas exchanges, and hope to address this in future work.

4. In Fig. 4. the CO2/H2O decoupling graph is not very clear. Please improve the illustration/graph/labelling to highlight the argument and link it with the examples given.

We agree that Fig. 4 was not adequately explained, and propose to add the following text at the end of the legend: "Line sinuosity depicts the percentage of transport that is diffusive." Also, to link this figure with its justification, we propose:

- a. to note in the legend that its justification is provided in the Appendix; and
- b. to add a final sentence to the Appendix stating that the consequences of Equation A5 are visualized in Figure 4, motivated also by the comments by Dr. Yakir.

This Viewpoint paper offers a provocative perspective of the role of water vapor in atmospheric and leaf-scale gas exchange. It argues that water vapor dynamics (WVD) can be a driver of gas transport phenomena, using first-principles reasoning and thought experiments. The paper challenges the conventional (and useful) practice that expresses gas concentrations relative to dry air, especially under very humid conditions. It raises valid conceptual challenges to current modeling frameworks in both atmospheric chemistry and plant ecophysiology.

Again, we thank the reviewer for this considered assessment of our preprint and for the suggestions that we believe will improve its clarity.

---

## Author Response (AR1)

Dear Dr. Merbold,

We thank you for your effort in handling this submission. Please find below our point-by-point replies (normal font) to your comments (bold italics), and in this re-submission also a marked-up manuscript version showing the changes that we have made as a consequence.

Sincerely,

Andrew S. Kowalski

***Replies to editor***

***your letter "Water vapour dynamics as a key determinant of atmospheric composition and transport mechanisms" has been seen by two reviewers of which both think the study can be a valuable contribution to our understanding and shortcomings in biosphere-atmosphere exchange measurements. I share this overall opinion.***

We thank the editor for this positive assessment.

***At the same time and thats less about the wording "air is water vapour" - which somehow got the attention of both reviewers similarly - I suggest some revisions to the current manuscript. I acknowledge the responses you have given concerning the ciritical points raised by the reviewers. Still the simple argument - we only focus on the physics here and will touch on the plant physiology later is too short. After all, the journals name and focus is on biogeosciences and not atmospheric chemistry and physics.***

We have made the recommended additions (see below for details).

***Still, I am confident that your letter will have a good position in biogeosciences with two additional adjustment (besides the suggestions made by the reviewers):***

***1) you already cite the relevant literature on where the proposed decoupling occurs - thus please include this, and***

We have expanded upon this with a mini-review of the literature. To try to maintain a readable text, we re-ordered the paragraph following Fig. 4 and broke it into two paragraphs, now found between lines 176-195 (line numbers without track changes).

***2) you already provide a good link to the real world examples in Figure 1 in the response to D. Yakir. Including this will only make your letter stronger and won't stop you from publishing an additional in -depth data driven paper at a later stage.***
***Thus, I am willing to accept this letter following the individual points raised and look forward to receiving the revised manuscript.***

We have now added both a new paragraph (now at lines 196-201) and the requested figure.